# Feasibility and Acceptability of Using Ecological Momentary Assessment to Evaluate Alcohol Use with American Indian Women

**DOI:** 10.3390/ijerph20126071

**Published:** 2023-06-07

**Authors:** Jessica D. Hanson, Amy Harris, Rebecca J. Gilbertson, Megan Charboneau, Marcia O’Leary

**Affiliations:** 1Department of Applied Human Sciences, University of Minnesota Duluth, 1216 Ordean Court, Duluth, MN 55812, USA; harr2542@d.umn.edu; 2Department of Psychology, University of Minnesota Duluth, 1216 Ordean Court, Duluth, MN 55812, USA; gilbertr@d.umn.edu; 3Missouri Breaks Industries Research Inc, 118 Willow Street, Eagle Butte, SD 57625, USAmarcia.oleary@mbiri.com (M.O.)

**Keywords:** ecological momentary assessments, alcohol, American Indian women

## Abstract

Background: Ecological momentary assessments (EMA) are one way to collect timely and accurate alcohol use data, as they involve signaling participants via cell phones to report on daily behaviors in real-time and in a participant’s natural environment. EMA has never been used with American Indian populations to evaluate alcohol consumption. The purpose of this project was to determine the feasibility and acceptability of EMA for American Indian women. Methods: Eligible participants were American Indian women between the ages of 18 and 44 who were not pregnant and had consumed more than one drink within the past month. All participants received a TracFone and weekly automated messages. Self-reported measures of daily quantity and frequency of alcohol consumption, alcohol type, and context were assessed once per week for four weeks. Baseline measurements also included the Drinking Motives Questionnaire-Revised (DMQ-R) and the Interpersonal Support Evaluation List (ISEL). Results: Fifteen participants were enrolled in the study. All but one participant completed all data collection time points, and drinking patterns were consistent across the study period. A total of 420 records were completed across 86 drinking days and 334 non-drinking days. Participants reported drinking an average of 5.7 days over the 30-day period and typically consumed 3.99 drinks per drinking occasion. Sixty-six percent of participants met gender-specific cut-points for heavy episodic drinking, with an average of 2.46 binge drinking occasions across the four week study period. Conclusions: This proof-of-concept project showed that EMA was both feasible and acceptable for collecting alcohol data from American Indian women. Additional studies are necessary to fully implement EMA with American Indian women to better understand the drinking motives, contexts, patterns, and risk factors in this population.

## 1. Background

Reducing alcohol consumption to below risky levels is a focus of Healthy People 2030, which sets data-driven national objectives to improve health in the United States (U.S.). Risky drinking is defined as five or more drinks at a time for men and 15 or more drinks per week. For women, risky drinking includes either binge (i.e., four or more drinks on an occasion) or heavy (i.e., eight or more drinks per week) drinking and any alcohol consumption during pregnancy [1]. Before the COVID-19 pandemic, national surveys found that 47.5% of females aged 12 or older were current drinkers [2], and approximately 20% of women aged 18 and older had at least one heavy drinking day in the past year [3]. Alcohol use in women is becoming more of a concern as rates of risky drinking rise. According to Karriker-Jaffe et al. (2018), rates of at-risk drinking in women rose from a rate of 18.4% in 2000 to 27.6% in 2010 [4].

Western countries, in particular, are seeing rising rates of alcohol-related harm in women [5]. This increase in risky drinking is especially concerning for women, as they are more likely to develop medical problems because of their drinking when compared to men [6]. For example, women who engage in risky drinking have higher risks of developing physical comorbidities when compared to men, such as alcohol-attributable cardiovascular diseases and diabetes [7], alcohol-related liver problems [8,9], alcohol-induced cardiomyopathy [10,11,12], and certain types of cancer [13]. As well, women face more psychiatric comorbidities, such as anxiety and depressive symptoms, when compared to men [14,15,16,17,18,19]. Finally, when compared to men, some women may increase their overall rate of alcohol consumption more quickly (e.g., from regular alcohol use to intoxication) and have a shorter amount of time before needing treatment for alcoholism, called “telescoping”, which may be due in part to entering treatment programs earlier than men [20,21,22,23]. 

Indigenous communities in the U.S. have faced particular scrutiny in regards to drinking alcohol while neglecting broader sociocultural and historical factors that may be related to substance use [24,25]. A study of American Indian women focused on drinking before and during pregnancy found that this population drank less before pregnancy when compared to white women, which “challenges commonly held beliefs of elevated alcohol consumption among American Indians compared with other races” [26]. Other recent findings conclude that American Indian women are more likely to abstain from alcohol compared to other racial/ethnic groups, although when they drink, they are more likely to binge drink when compared to non-Native communities [27]. Other studies in general samples of American Indian groups have similar findings, with many American Indian/Alaska Native groups exhibiting lower rates of substance abuse when compared to the U.S. general population, including higher rates of complete alcohol abstinence and significant variability within American Indian/Alaska Native tribes [28,29,30,31]. As Whitesell et al. (2012) state, “It is clear that substance use and disorder threaten the health of AI/AN communities, yet neither are universally pervasive within these communities” [31].

Much of what is known about drinking in American Indian women comes from retrospective data collection, such as through utilization of the Timeline Followback [27,32], surveying individuals about past binge drinking [33,34], and through qualitative questions, such as establishing the specific types of alcohol consumed, size of containers, number of individuals sharing, and timeframe of drinking episodes [27,33]. However, these methods do not always capture predictors or clear patterns of drinking, which can be important in developing interventions [35,36]. As well, there are few research studies that focus on non-drinking days or protective factors against risky drinking in adult American Indian women, which could ultimately be used in intervention development. Therefore, more data on the contextual variables, including type of alcohol consumed, place where alcohol was consumed, number of people in the drinking group, timing of drinking, and other psycho-social variables such as mood and the affect surrounding a drinking episode [37,38], is necessary to better understand drinking patterns in this population and to subsequently inform interventions to reduce risky drinking in this population. 

One solution is the use of ecological momentary assessment (EMA), which involves signaling participants via cell phones or other devices multiple times per day, for a period of days or weeks, to report on daily behaviors in real-time and in a participant’s natural environment [39]. These reports include the associated psychological states and/or environmental conditions that surround behaviors. EMA has been utilized in a variety of health fields, including chronic pain, weight loss, mood, depression, and stress, smoking, eating disorders, and general health promotion/disease prevention [40,41,42,43,44,45], as well as in a variety of age groups [46]. As Shiffman (2009) notes, EMA works well for substance use because it is focused on collecting event-level data on behavior that is often related to mood and context [36]. EMA reduces retrospective recall bias, which provides better data on binge drinking patterns (the type of alcohol, precipitants of drinking, and the situations in which the risky drinking is occurring) and also allows for data collection on non-drinking days to potentially identify protective factors against drinking [36]. 

EMA has been extensively used to better understand drinking behaviors among general populations [47,48], including several studies that used EMA with college students to better understand binge drinking patterns and co-occurring behaviors, such as smoking and alcohol use [49,50,51,52,53,54,55,56]. This approach has also been used with high-risk populations, such as women with depression, to understand the role of affect on drinking patterns [57]. EMA has also been used to understand the impact of alcohol use, such as in research on hangovers and alcohol-impaired driving [58,59,60,61]. There are two major systematic reviews on the use of EMA to understand alcohol behaviors [36,62], with the most recent finding being 175 studies on alcohol and EMA [62]. 

EMA has not previously been utilized with American Indian women to assess drinking but has great potential for resolving some of the concerns cited above, including issues with retrospective data collection, unclear data on contexts of drinking and non-drinking days, and data focusing primarily on high-risk groups rather than drinking in general populations. The use of EMA with American Indian women living in rural and reservation communities is novel, and it could be a useful tool in better understanding drinking patterns using real-time data collection. Therefore, the goal of the study described here was to utilize EMA in a small proof-of-concept study as a way to establish the feasibility and acceptability of EMA with American Indian women and to collect alcohol data using an electronic data capture system. 

## 2. Materials and Methods

### 2.1. Eligibility and Recruitment

Institutional review board approval was obtained at both the tribal level and at the principal investigator’s institution (STUDY00018669). Eligible participants were American Indian women from one Great Plains tribe who were 18 to 44 years of age, were not pregnant, and reported consuming one or more alcoholic drinks within the past month. A local project coordinator was hired to recruit and enroll participants, and they disseminated flyers to recruit participants. Interested individuals called the Project Coordinator, who utilized a script to determine eligibility and complete verbal consents. Verbal consents were sought because data collection occurred during the COVID-19 pandemic; therefore, in-person contact was discouraged. Additionally, the project used verbal consent because a signed consent would have been the only document where a participant’s name would have been listed. Recruitment continued until there were 15 eligible participants. 

### 2.2. Data Collection

Once consented to, all participants received a Tracfone with a unique telephone number that was used for the study. A Tracfone is a prepaid, no-contract mobile phone. This phone included a 30-day phone plan with a predetermined number of minutes and mobile web access with nationwide coverage. When the Tracfones were dropped off at participants’ homes, the project coordinator spent 5–10 minutes explaining the data collection system and gave each participant a handout with similar information. Participants also received the project coordinator’s business card, so they could call if they had problems or questions during data collection. 

Next, the participant’s Tracfone phone number was forwarded to the lead investigator and entered into the ReTAINE® (Real Time Assessment in the Natural Environment) system, an automated internet-based system for designing, administering, and managing EMA. The ReTAINE system utilizes automated prompts sent to the participants’ cell phones via text or email, where participants click on a link to complete a web-based assessment [43]. No additional identifiable information from the participants was shared with the study team or displayed in the patient signaling system.

Using ReTAINE, participants were contacted for data collection once per week over the course of four weeks. Self-reported measures of alcohol consumption, alcohol type, and context were assessed once per week for four weeks (e.g., baseline/week 1, weeks 2–4). Baseline/week 1 measurements also included the Drinking Motives Questionnaire-Revised (DMQR) and the Interpersonal Support Evaluation List (ISEL). Each weekly survey took an estimated 5–10 minutes to complete. Approximately 1–2 days after completing the week 4 data collection, participants were additionally asked to provide their opinions on the study overall, survey questions, and the use of EMA. If there was no answer to the prompt, at least one reminder text was sent by the project coordinator. Participants stopped receiving texts if they did not complete the first week’s survey or if they missed two consecutive weeks of surveys, although data was still included in the final analysis if available. After data collection, the participants were not contacted again and were allowed to keep the TracFones.

### 2.3. Measurements

To determine appropriate measures for this project, we utilized the methods and expertise of researchers with previous experience both with EMA and with using EMA to collect alcohol data [37,38,48,58]. Based on recommendations from previous research and from our community partners, EMA measures were kept brief to reduce participant burden [63]. Alcohol consumption was measured by three questions about drinking per week, adapted from previous research [57]: (1) *In the last week, how often have you had any kind of beverage containing alcohol, whether it was beer, a wine cooler, shots, or any other alcoholic drink?* (categorical: everyday, 5–6 days per week, 3–4 days per week, 1–2 days per week, or none at all); (2) *For each day you drank in the last week, how many drinks do you estimate you had each day?* (continuous: number of drinks, 0+); and (3) *In the last week, how often did you have 4 or more drinks in a two-hour period?* (categorical: everyday, 5–6 days per week, 3–4 days per week, 1–2 days per week, or none at all). Weekly alcohol measurements also included *what* participants typically drank when they drank alcohol, where they were asked to check the type of alcohol they usually drank (categorical: various types of alcohol listed based on previous formative research with tribal communities [64]), and *how* they drank (categorical: in a group, at my or someone else’s home; alone, at home; in a group, at a bar or another public place; alone, at a bar or another public place). The alcohol consumption questions were asked during all four weeks of data collection. 

Baseline measurements also included the DMQ-R and the ISEL-shortened version. The DMQ-R measures how people score on a five-point scale to questions on four motivational dimensions: social motives (drinking to be sociable, to celebrate parties); coping motives (drinking because it makes one forget their problems); enhancement motives (drinking to feel better or to be able to do things otherwise impossible); and social pressure and conformity motives (drinking because others do, to fit in) [65,66,67]. The ISEL-shortened version was designed to measure perceptions of social support and is a four-point scale of agreement regarding statements concerning the perceived availability of potential social resources (appraisal support, belonging support, and tangible support) [68].

Immediately following the week 4 alcohol data collection, participants were asked to provide feedback as a way to evaluate the feasibility and acceptability of EMA. Questions were adapted from previous research [69,70,71] and included a five-point scale of agreement on three major sections: the questions overall (e.g., “I understood the questions”), the alcohol variables specifically (e.g., “I accurately recorded my alcohol use”), and overall participation in the study (“It was easy participating in this study”). A full list of questions is included in the Results, Table 1.

### 2.4. Data Analysis

Data were analyzed using SAS Version 9.4 (SAS Institute, Inc., Cary, NC, USA) and merged for each participant across the four weeks of project data collection. Continuous drinking data, collected for each day of the week since the last survey, was used to create alcohol consumption measures. A new variable, “quantity”, was calculated by summing the number of drinks for each week and total per project period. The quantity calculation included the beverage type (i.e., beer, flavored malt beverage, wine, or hard liquor) to ascertain the standard drink. Drinks per occasion were then calculated as the total drinks divided by the number of total drinking occasions. Participants reporting four or more drinks on a single occasion during the study period were considered binge drinkers. The number of binge drinking (e.g., greater than or equal to four drinks in a single setting) occasions was also summed for a single participant. 

A second new variable—“frequency”—was calculated through the number of times per week that alcohol was consumed and averaged to ascertain the number of days, on average, participants reported drinking across the study period. The frequency of drinking per week was compared to the drinking data reported per day, once per week, to ascertain the consistency of reporting by participants. In addition, using the quantity and frequency variables, a quantity and frequency index (QFI) [72] was calculated for each participant for each week. For example, someone consuming one 12-ounce beer per day, each day of the week, would score a QFI of 60. An average QFI was also calculated at the end of the study across the four weeks of data collection. If a zero was reported, it was designated as a non-drinking day.

Categorical drinking data, including drink type (what participants drank) and setting (how participants drank, dichotomously coded as either “alone” or “group”), as well as responses to the feedback survey questions, were aggregated and averaged using basic descriptive statistics. Following, the data were analyzed via chi-square to assess the association between context and the prevalence of binge drinking among participants. Finally, the scale scores for the DMQ-R and ISEL were calculated as the sum of the respective items on each scale for both questionnaires. Following, Pearson r correlation coefficients were used to assess the association between DMQ-R subscales, ISEL subscales, and drinking variables.

## 3. Results

The data were collected once per week for four weeks. Fourteen of the fifteen participants (93%) completed all four weeks of data collection, and one participant completed three out of the four weeks of data collection, not responding to the final week of data collection as she gave her project phone to a family member. Thus, participants carried the EMA for 27.5 days (SD = 1.8), on average, and a total of 413 records were completed across 86 drinking days and 327 non-drinking days. Drinking patterns were consistent across the study period: Week 1 drinks were positively related to drinks at week 4, *r* = 0.70, *p* =0.003. Participants’ consistency in reporting was further assessed by comparing continuous data collected for each day of the week (number of days that drinking was recorded) with the categorical frequency question (how many days in the last week did you consume alcohol). Across the study period, concordance in frequency reporting was 84%, or 50 out of 59 records. 

For the quantity variable, when participants drank, they consumed an average of 3.99 drinks per drinking occasion (SD = 3.1; range 1–13). The number of drinks each week ranged from 4.7 (Week 3 data collection) to 6.8 (Week 4 data collection), with an average total amount of 22.3 (SD = 21.5; range 0–84) drinks over the course of the entire four weeks of data collection. For the frequency variable, participants reported drinking an average of 5.7 days total over the four-week period (SD = 3.3; range 0–12). The number of days that participants drank per week ranged from 1.2 (week 4 data collection) to 1.5 (weeks 1, 2, and 3 data collection). There was one participant who reported not drinking during the four weeks. Without this participant included, the QFI average for all four weeks of data collection was 0.3 (SD = 0.3; range 0–1.14).

The majority of participants (66.7%) met gender-specific cut-points for at least one binge drinking episode, with an average of 2.5 binge drinking occasions (SD = 3.1) within the four-week study. Binge drinking was marginally more likely to occur in a group versus drinking alone, χ^2^ (n = 15) = 3.73, *p* = 0.0534. Likewise, binge drinking was more likely to occur in a bar versus at home, χ^2^ (n = 15) = 10.81, *p* < 0.001. The most common type of alcohol consumed was beer (50.0% of reports across 4 weeks), followed by flavored malt alcohol (15.5%), wine (15.5%), liquor that was shared with others (13.8%), and liquor that was consumed alone (1.7%). Overall, participants drinking with a group were more likely to consume alcohol at a bar (49.2%), compared to drinking alone at home (27.1%), χ^2^ (n = 15) = 24.59, *p* < 0.001. 

Of the four subscales of the DMQ-R—social motives, coping motives, enhancement motives, and social pressure/conformity motives—the social pressure/conformity motive showed the highest group mean value with an average of 11.6 (SD = 3.7; range 8–21), with the coping motive showing the lowest group mean value with an average of 10.3 (SD = 4.3; range 6–21). The enhancement motive was marginally positively associated with the number of drinks per occasion, *p* = 0.09. Of the three subscales of the ISEL—tangible support, appraisal support, and belonging support—appraisal support showed the highest group mean value with an average of 13.2 (SD = 2.7; range 7–16). In addition, the tangible subscale was found to be negatively related to responses to all drinking motive subscales of the DMQ-R, specifically the social motives subscale (*p* = 0.04), the coping motive subscale (*p* = 0.05), the enhancement motive subscale (*p* < 0.01), and the social pressure/conformity motive subscale (*p* < 0.01). However, social support was not associated with the quantity and frequency of drinking variables in this study.

In addition to being feasible for alcohol use data collection with this population, EMA also appeared to be acceptable to the sample. Based on feedback from the post-study survey, participants appeared to agree that EMA, especially this method for collecting data on alcohol, was suitable, and they acknowledged the benefits of participating in the study. See Table 1 for the average level of agreement with statements regarding participation and use of EMA. 

**Table 1 ijerph-20-06071-t001:** Participant-reported feasibility/acceptability.

Statement Regarding Participation	Average Level of Agreement
**Overall Acceptability of EMA**	
I understood the questions.	4.92
I could figure out how to answer the questions on my phone.	4.77
The questions were relevant to me and my situation.	4.08
Completing the questions was convenient.	4.31
Completing the questions was easy.	4.92
Completing the questions was challenging.	1.31
Completing the questions was stressful.	1.54
Completing the questions interfered with other activities.	1.23
**Acceptability of EMA with Alcohol**	
Answering questions about alcohol use made me feel down or discouraged.	1.69
I did my best to answer all the questions in a timely manner.	4.23
I accurately recorded my alcohol use.	4.38
I answered all the questions about my alcohol use.	4.92
Answering questions about my alcohol use made me really think about my drinking.	3.85
Answering questions about my alcohol use helped me decrease my drinking.	3.46
The questions were asked in a way to accurately measure my drinking.	3.92
**Acceptability of Participation**	
I enjoyed participating in this study.	4.69
It was easy participating in this study.	4.85
I would be willing to participate in a similar study in the future.	4.62
I would have been willing to participate without the phone card or TracFone.	3.77
I could have participated without the phone card or TracFone.	4.23
I would recommend a similar study to a friend.	4.62

Scale responses: 1 = Strongly disagree; 2 = Disagree; 3 = Neutral; 4 = Agree; 5 = Strongly agree.

## 4. Discussion

EMA has been extensively used to understand drinking behavior and consequences, with a recent review article finding that there have been 175 published articles on the topic [62]. However, this methodology has never before been used with American Indian populations to understand their particular drinking patterns. While American Indian women are more likely to abstain from alcohol compared to other racial/ethnic groups, those who drink do so at risky levels (e.g., very few exhibit low to moderate drinking patterns) [27]. Preliminary data from previous studies show that American Indian women living in reservation communities tend to drink with groups of people in a home setting rather than in bars and restaurants, and that reasons for binge drinking include to be sociable, to celebrate a special occasion, because others are drinking, or to be part of a group [33,34]. However, more data on the situational context is necessary to better understand drinking patterns in this population and, subsequently, inform interventions to reduce risky drinking. EMA can help provide better data on binge drinking patterns (the type of alcohol, precipitants of drinking, and situations in which the risky drinking is occurring) [73]. By assessing subjects in their natural environments, EMA provides a natural range and variation of responses, and “generalization to the real world is built in” [74].

The study described above was considered a proof-of-concept study, similar to an early stage of a clinical trial, where a small number of participants—often those who are accessible rather than representative—receive the “treatment” to better inform the “dose” and method of delivery [75]. These types of small studies focus on determining the essential features of a new intervention or how to adapt an existing treatment for further testing. In this phase, protocols are fluid, and it is expected that there will be ongoing, iterative adjustments in response to evolving findings. Ultimately, early studies are used to determine if the “treatment” merits more rigorous testing in future efficacy trials [75]. In the case of the study described above, the goal was to see if and how EMA could work in a rural, reservation-based community, including the essential tools necessary to implement this type of data collection on a larger scale. While not employing typical EMA data collection, the design of the proof-of-concept study described here helped to establish the potential of EMA to understand the drinking patterns and the social context of drinking among American Indian women. 

Such proof-of-concept strategies are valuable when considering the history of research with American Indian communities, particularly alcohol research, which often perpetuates harmful stereotypes regarding drinking in this population [76]. Establishing feasibility and acceptability by directly engaging communities is an essential first step in developing research questions and collecting data locally. We followed community engagement principles by working with a local, American Indian-owned business to set up the project, training an enrolled tribal member to consent participants and collect data, and having established regular check-ins to ensure fidelity and quality data collection.

Therefore, the first aim of this study was to establish the feasibility of EMA in rural American Indian communities, using alcohol consumption as the target behavior. Previous anecdotal information received by this team indicated a concern with the use of technology to collect data in rural and reservation communities. However, this did not prove to be the case for this EMA project. We maintained high retention in our proof-of-concept study over the four weeks of data collection. Out of the 15 participants, all but one completed all data collection points, and the one participant who did not complete all data points still completed data collection for three out of the four weeks. Several participants needed reminders to complete the data collection, and a small number needed the survey sent to them, which also aided in high retention. Of additional note in regards to the feasibility of EMA with rural/reservation American Indian women was the high response rate in spite of the COVID-19 pandemic and the generally consistent data collection from participants. 

The project also assisted in highlighting the importance of providing equipment and training to participants. Budgeting for study-specific hardware (e.g., Straight Talk phones and cell phone minutes) for participants at no cost to them aided in successful remote data collection with American Indian women. Only one participant no longer had the project phone at the end of data collection because she had given the phone to a relative for their use; there were no other issues with loss of equipment. In addition, having a local staff person hired specifically for the project for recruitment and enrollment was essential. This individual was able to utilize community contacts and current relationships to aid in recruitment and was the main contact for all project-related questions. Local staff were particularly helpful in training individuals on the system, connecting via text with them for retention purposes, and answering questions about the program via email and telephone.

The amount and quality of data also highlight the feasibility of EMA in studying alcohol use in American Indian communities. There were few missing data points, and drinking patterns were consistent across the study period (e.g., week 1 drinking reports were positively correlated with drinks at week 4). Data from this study is consistent with other studies on drinking with Native women [27]: participants were more likely to abstain from alcohol during the four-week period than drink regularly, but when they did drink, they binged and had an average of over two binge drinking occasions within the entirety of the study. Additionally, as with other studies, drinking, particularly binge drinking, typically happens in a group setting. However, unlike previous studies, much of the binge drinking occurred in public spaces, such as bars, rather than in private homes [33,34]. Drinking motives appeared to be associated with the quantity and frequency of alcohol use, although social support did not appear to play as much of a role in drinking behaviors. Additional research is needed to understand these potential relationships more.

The second aim of the study was to determine the acceptability of EMA collecting alcohol data from American Indian women. Responses to the post-data collection survey indicated that EMA was acceptable to the population. Overall, participants agreed that EMA, especially this method for collecting data on alcohol, was suitable, and they acknowledged the benefits of participating in the study. Overall, participants seemed to agree that EMA was acceptable, and they understood the questions that were asked in the weekly survey (Table 1). Respondents agreed that EMA use for alcohol use was acceptable and appeared to accurately record their alcohol use. The majority of participants reported enjoying their participation in the study, agreed that the study was easy to complete, and stated they would participate in a future similar study.

## 5. Limitations

There were some limitations to this study. It was a small sample size with one tribal community, although this fits with the goal of the proof-of-concept approach to establish feasibility and acceptability before developing a larger study with multiple communities where demographics will be collected. The project provided the “equipment” (e.g., cell phones) to complete the study instead of having participants use their own phones, which may be cost prohibitive in larger samples but did aid in consistency during recruitment and likely promoted retention. Finally, we collected data once per week rather than multiple times per day, as is typical in an EMA study. Again, this was the chosen methodology because of the desire to test the system with our sample rather than focus on the behavior itself. Regardless of our modified EMA study with the small sample, we were still able to collect significant data that warrants additional focus via a traditional EMA study.

## 6. Conclusions

Ultimately, EMA is an additional tool to better understand drinking in American Indian communities. In general, EMA works well for substance use because it is focused on collecting event-level data on behavior that is often related to mood and context, and it also reduces the risk of retrospective recall bias, which provides better data on drinking (e.g., type of alcohol, situations in which drinking is occurring) [36]. As was shown in our study, we were able to include data on non-drinking days instead of focusing just on drinking episodes; larger EMA studies of this type can better identify protective factors against drinking. EMA is a feasible way to collect substance use data with American Indian communities, although the outcomes of a larger, more rigorous study will provide a more traditional use of EMA with American Indian communities. 

## Data Availability

Data are not publicly available. Tribes, as sovereign nations, own and control any data collected and have a specific process for requesting permission to access the data.

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
