# Peer review of "Feasibility and Acceptability of Using Ecological Momentary Assessment to Evaluate Alcohol Use with American Indian Women"

_ijerph, 2023, doi:10.3390/ijerph20126071_

Round 1

Reviewer 1 Report

Thank you for the opportunity to review this article.

line 10 ? alcohol use data - this sentence and the next is not clear and could do with being developed to explain.

Maybe define what EMA which will help make this clearer - are these the 5 surveys mentioned in line 18 - as this is not clear and how this relates together

reference technique not correct

also an issue with women is fetal alcohol spectrum disorder

line 89 - discuss non drinking days and why there is benefits to this

has EMA been shown to be reliable and accurate

I assume the surveys were loaded onto the phone - not clear

no mention of ethics approval

line 262 'n' ? meant to be there or is a typo

line 264 end of this sentence is not most common alcohol consumed. Make new sentence

some repetition in this paragraph

line 291 remove one of 'study'. This sentence is not clear

line 296 - should be 'it is'

some yellow highlighting that should not be there

recommendations needed

Line 14 what is TracFone and how does it relate

Author Response

  • line 10 ? alcohol use data - this sentence and the next is not clear and could do with being developed to explain.

We added the word “use” in line 10 to clarify this sentence. We also added prose in lines 10-11 to help explain the subsequent sentence.

  • Maybe define what EMA which will help make this clearer - are these the 5 surveys mentioned in line 18 - as this is not clear and how this relates together

We clarified this sentence in line 20 of the Track Changes version. 

  • reference technique not correct

The references have been fixed throughout the document.

  • also an issue with women is fetal alcohol spectrum disorder

We are happy to address this comment, but are not clear what this is referring to in the text.

  • line 89 - discuss non drinking days and why there is benefits to this

As noted in lines 95-96, the benefit of having information about non-drinking days is to potentially identify protective factors against drinking.

  • has EMA been shown to be reliable and accurate

Yes, EMA has been utilized in a variety of health fields, including chronic pain, weight loss, mood, depression, and stress, smoking, eating disorders, and general health promotion/disease prevention, as well as in a variety of age groups. EMA has also been used extensively in alcohol research. See lines 98-108 in the Track Changes version for specific citations. 

  • I assume the surveys were loaded onto the phone - not clear

Yes, as noted in lines 143-144 of the Track Changes version, the data collected system we used utilized automated prompts to the participants’ cell phone via text or email, where participants click on a link to complete a web-based assessment.

  • no mention of ethics approval

As specified in lines 118-119 of the Track Changes version, institutional review board approval was obtained at both the tribal level and at the principal investigator’s institution. We added the ethic’s approval number on line 119 of the Track Changes version. 

  • line 262 'n' ? meant to be there or is a typo

Line 262 is line 256 in the Track Changes version. Yes, this ‘n’ is meant to be there to signify the number of participants (e.g., small n, or small number of participants). 

  • line 264 end of this sentence is not most common alcohol consumed. Make new sentence

We added “that was” in lines 259 and 260 of the Track Changes version to help clarify that we had 2 separate variables regarding liquor - individuals could state that they drank liquor with others, and individuals could state that they drank liquor alone.

  • some repetition in this paragraph

We agree and made edits to lines 260-265 of the Track Changes version. 

  • line 291 remove one of 'study'. This sentence is not clear

Please see the edit on line 305 of the track changes version.

  • line 296 - should be 'it is'

Please see the edit on line 310 of the Track Changes version.

  • some yellow highlighting that should not be there

We removed all highlighting from the document.

  • recommendations needed

Please see prose in lines 311-312 (e.g., “...early studies are used to determine if the “treatment” merits more rigorous testing in future efficacy trials”), and lines 387-388 (e.g., “...larger EMA studies of this type can better identify protective factors against drinking. EMA is a feasible way to collect substance use data with American Indian communities, although the outcomes of a larger, more rigorous study will provide a more traditional use of EMA with American Indian communities.”) 

  • Line 14 what is TracFone and how does it relate

We added a definition of a Tracfone on line 132 of the Track Changes version. 

Reviewer 2 Report

The authors should provide information on their use of ‘American Indian women’. Is this the accepted way to refer to this group in the literature? Did the participants agree that this was the label they would prefer?

“…47.5% of females aged 12 or older were current drinkers …and approximately 20% of women aged 18 and older had at least one drinking day in the past year”

I might be missing something, but if 47.5% are current drinkers (which I assume is a much shorter time span than a year), then would the % women that had a drink in the past year not be similar or higher than this?

There should be a more thorough review of the use of EMA for alcohol consumption. The authors provide just two references when stating that EMA has been used to understand drinking in general populations. They provide no references when stating that EMA might have utility when focusing on high-risk groups. With respect to high-risk groups, EMA has been employed with student populations (Riordan et al., 2015). There is also a broader literature on the use of EMA for mood and affect measurement that might be relevant to questions about the contexts of drinking.

Given the current study 1) focuses on American Indian women and, 2) highlights issues with previous methods of data collection and interpretation of data with this population, more information must be given about how the current study was developed. Do any of the authors identify as American Indian? If not, this represents a serious issue, especially if there was little consultation with American Indian women when designing the study (Smith, 2013).

Unless I missed it, there is no information on the average age of participants or any related demographic information (e.g., level of education, number of children, etc.). These variables are essential, as many relate to alcohol consumption.

What is the justification for the sample size of 15? Is this based on power analysis or any other method?

In the Discussion section, is there a need to mention that a phone was not returned? The Discussion also makes it seem like there was little, if any, community contact by the authors themselves. The authors need to elaborate on their approach. If the purpose of the study is to develop approaches to collect more reliable data from American Indian women, then it would seem that part of the approach should be connection and consultation by the authors with that community.

Riordan, B. C., Scarf, D., & Conner, T. S. (2015). Is orientation week a gateway to persistent alcohol use in university students? A preliminary investigation. Journal of Studies on Alcohol and Drugs, 76, 204-2011. https://doi.org/10.15288/jsad.2015.76.204

Smith, L. T. (2013). Decolonizing methodologies: Research and indigenous peoples. Zed Book.

Author Response

  • The authors should provide information on their use of ‘American Indian women’. Is this the accepted way to refer to this group in the literature? Did the participants agree that this was the label they would prefer?

Yes, this is a common way to refer to this population in the literature. The first author of this manuscript has worked with tribal communities for many years and has collaborated on different papers and presentations where this term is used. Other researchers and authors use this term to describe the population in other studies. Finally, two of our co-authors are tribal members and/or work for a tribal organization and approve this language to describe the population. 

  • “…47.5% of females aged 12 or older were current drinkers …and approximately 20% of women aged 18 and older had at least one drinking day in the past year” I might be missing something, but if 47.5% are current drinkers (which I assume is a much shorter time span than a year), then would the % women that had a drink in the past year not be similar or higher than this?

We clarified that the 20% is specific to “heavy drinking day” in line 39 of the Track Changes version.

  • There should be a more thorough review of the use of EMA for alcohol consumption. The authors provide just two references when stating that EMA has been used to understand drinking in general populations. They provide no references when stating that EMA might have utility when focusing on high-risk groups. With respect to high-risk groups, EMA has been employed with student populations (Riordan et al., 2015). There is also a broader literature on the use of EMA for mood and affect measurement that might be relevant to questions about the contexts of drinking.

Additional prose on use of EMA with various populations has been added to the background, beginning on line 98 in the Track Changes version. This includes the addition of cited literature on the use of EMA with college students’ drinking and the use of EMA to better understand the role of affect on drinking.

  • Given the current study 1) focuses on American Indian women and, 2) highlights issues with previous methods of data collection and interpretation of data with this population, more information must be given about how the current study was developed. Do any of the authors identify as American Indian? If not, this represents a serious issue, especially if there was little consultation with American Indian women when designing the study (Smith, 2013).

Yes, one of the co-authors is an enrolled member of the tribe this data was collected with.

  • Unless I missed it, there is no information on the average age of participants or any related demographic information (e.g., level of education, number of children, etc.). These variables are essential, as many relate to alcohol consumption.

We did not collect demographic information beyond our eligibility criteria (American Indian women from one Great Plains tribe who were 18 to 44 years of age, who were not pregnant, and who reported consuming one or more alcoholic drinks within the past month). We recognize this as a limitation of this study and added information in the Limitations section, line 380 of the Track Changes version. 

  • What is the justification for the sample size of 15? Is this based on power analysis or any other method?

The n = 15 is not based on a power analysis. Because the goal was to see if and how EMA could work in a rural, reservation-based community, we chose 15 as our sample size based on the small budget we had for this project. The small sample size is recognized in the Limitations section.

  • In the Discussion section, is there a need to mention that a phone was not returned? 

We feel this is important to note for other researchers who may purchase phones for an EMA study - we did not have any major issues with participants losing project materials and would like that noted.

  • The Discussion also makes it seem like there was little, if any, community contact by the authors themselves. The authors need to elaborate on their approach. If the purpose of the study is to develop approaches to collect more reliable data from American Indian women, then it would seem that part of the approach should be connection and consultation by the authors with that community.

We expounded upon our community engagement by adding a paragraph starting in line 319 of the Track Change version. 

The project builds on previous work by a study author (J.H.) who has worked with tribes for over fifteen years to adapt, implement, and evaluate alcohol-exposed pregnancy interventions. This included a feasibility and acceptability study, a community needs assessment, and a reliability/ validity study, where J.H. and colleagues collaborated with tribes to complete a mixed methods approach to gather input into the alcohol survey measures from content experts, adult American Indian women of childbearing age, and via a test retest methodology. While not added to the paper itself, we are happy to add information on our long-standing relationships with tribal communities, such as this historical information.

Reviewer 3 Report

The study deals with a topic of interest such as the registration of alcohol consumption in American Indian Women in rural areas. I believe that the extrapolation of this methodology in other rural or minority communities can be discussed.

While it is true, the EMA has not been applied in "real" conditions; They are given the phone, which can greatly increase participation, and they are asked for information weekly, not daily, but the subject is interesting, as is the study.

I believe, however, that there are some aspects that should be modified or reviewed to improve the document;

In my opinion, the introduction is correct and well referenced.

1. As a better consideration, on some occasions the authors cite an author within the text and cite him again in the same line, they should review this aspect.

2. I miss explicit information regarding the approval of the ethics committee with the corresponding reference number

3. I am struck by the lack of written informed consent, I would appreciate it if the authors could explain the reason for this decision

4. I would like the authors to explain in more detail why they chose the questions regarding alcohol consumption, and indicate if they are using validated questions? Why did the authors select these questions?

5. I personally miss information regarding days of no alcohol use, What were the participants doing? were they alone? ... information that the authors explained as interest at the introduction, can i ask why the lack of this information?

6. In my opinion table 1 should be part of the results of the study

7. Results: I miss tables in which you can see the data described in results, beyond table 1, which in my opinion, should be in results

8. I believe that the discussion section is the one that should have the most changes, in my opinio there is an important lack of bibliography that supports or not the statements in the discussions section. Authors should enrich this section with scientific evidence

Author Response

  • As a better consideration, on some occasions the authors cite an author within the text and cite him again in the same line, they should review this aspect.

The references have been fixed throughout the document.

  • I miss explicit information regarding the approval of the ethics committee with the corresponding reference number

We added the ethic’s approval number on line 119 of the Track Changes version. 

  • I am struck by the lack of written informed consent, I would appreciate it if the authors could explain the reason for this decision.

Additional justification on our use of a verbal consent is now added to the prose, lines 124-128 of the Track Changes version. 

  • I would like the authors to explain in more detail why they chose the questions regarding alcohol consumption, and indicate if they are using validated questions? Why did the authors select these questions?

We've inserted language and references to provide additional rationale for the selection of questions asked to respondents; see lines 160-164 in the Track Changes version. 

  • I personally miss information regarding days of no alcohol use, What were the participants doing? were they alone? ... information that the authors explained as interest at the introduction, can i ask why the lack of this information?

We did not ask additional information if it was a non-drinking day in this feasibility/acceptability study. However, future studies with American Indian participants using EMA will ask for this information to better understand non-drinking days. 

  • In my opinion table 1 should be part of the results of the study

We agree with this and moved the table to the Results, line 285 of the Track Changes version. 

  • Results: I miss tables in which you can see the data described in results, beyond table 1, which in my opinion, should be in results

We agree with this and moved the table to the Results, line 285 of the Track Changes version. 

  • I believe that the discussion section is the one that should have the most changes, in my opinion there is an important lack of bibliography that supports or not the statements in the discussions section. Authors should enrich this section with scientific evidence

We added a paragraph in the beginning that helps support the subsequent statements in the Discussion. See this addition beginning in line 289 of the Track Changes version. 

Round 2

Reviewer 2 Report

Thank you for the responses. My main concerns were consultation and inclusion, and the authors have adequately addressed these points.